# Machine Learning Analysis for Phenolic Compound Monitoring Using a Mobile Phone-Based ECL Sensor

**DOI:** 10.3390/s21186004

**Published:** 2021-09-08

**Authors:** Joseph Taylor, Elmer Ccopa-Rivera, Solomon Kim, Reise Campbell, Rodney Summerscales, Hyun Kwon

**Affiliations:** 1School of Engineering, Andrews University, Berrien Springs, MI 49104, USA; tjoseph@andrews.edu (J.T.); ccoparivera@andrews.edu (E.C.-R.); 2Department of Computing, Andrews University, Berrien Springs, MI 49104, USA; ksolomon@andrews.edu (S.K.); reise@andrews.edu (R.C.); summersc@andrews.edu (R.S.)

**Keywords:** ECL, low-cost sensor, mobile phone-based sensor

## Abstract

Machine learning (ML) can be an appropriate approach to overcoming common problems associated with sensors for low-cost, point-of-care diagnostics, such as non-linearity, multidimensionality, sensor-to-sensor variations, presence of anomalies, and ambiguity in key features. This study proposes a novel approach based on ML algorithms (neural nets, Gaussian Process Regression, among others) to model the electrochemiluminescence (ECL) quenching mechanism of the [Ru(bpy)_3_]^2+^/TPrA system by phenolic compounds, thus allowing their detection and quantification. The relationships between the concentration of phenolic compounds and their effect on the ECL intensity and current data measured using a mobile phone-based ECL sensor is investigated. The ML regression tasks with a tri-layer neural net using minimally processed time series data showed better or comparable detection performance compared to the performance using extracted key features without extra preprocessing. Combined multimodal characteristics produced an 80% more enhanced performance with multilayer neural net algorithms than a single feature based-regression analysis. The results demonstrated that the ML could provide a robust analysis framework for sensor data with noises and variability. It demonstrates that ML strategies can play a crucial role in chemical or biosensor data analysis, providing a robust model by maximizing all the obtained information and integrating nonlinearity and sensor-to-sensor variations.

## 1. Introduction

Extensive progress has been made in chemical and biosensor technology to provide reproducible and reliable data in a fast and low-cost setting. These efforts include developing enhanced transducers, finding/engineering better binding moieties, devising signal amplification strategies, developing strict control of sensing conditions, and developing improved sensing strategies [1,2,3]. In contrast to such rapid development in hardware and sensing strategies, less attention has been paid to the data analysis of these sensors. It is important to provide consistent sensing conditions for quality control, but it results in a cost increase through rigorous hardware development, environment control, and automated operation. Thus, there arises a strong need for transform data analytics to address some of the challenges in low-cost sensor systems.

ML has emerged as a powerful tool for a wide range of sensor data analysis, including analyte detection, protein-protein interaction, wearable sensors, environmental pollutant monitoring, and sensor arrays [4,5,6,7,8,9,10]. ML can provide novel strategies for overcoming challenges faced by common sensors and detecting species or concentrations of analytes based on a trained algorithm. The advantages of using ML include the capability of detecting anomalies, noise reduction, categorizing signals, and most importantly, it can find unforeseen interrelations between signals and chemical- and/or bio-events in the sensor with advanced data-driven strategies [6]. Algorithms, such as support vector regression (SVR), decision trees, neural networks (NN), and Gaussian Process Regression (GPR) have been employed to sensor analysis in recent years [11]. ML applications for sensor data is crucial as there is a strong need for evaluating multidimensional and nonlinearities of the detected signals. Traditional data analysis for chemical and biosensors have relied heavily on calibration curves with a predetermined feature. Even under stringent control, it is hard to avoid sensor-to-sensor variations due to sensor unit replacements that would require re-calibration. ML can provide methods to overcome these challenges faced by those sensors and predict species or concentrations of analytes effectively [6,12].

The ECL sensing scheme has received significant attention as a platform of light-emitting sensors and an analytical detection method. Because ECL does not require any external excitation-light source, it has the advantage of having ultra-sensitivity, generating an exceptionally low background signal. In addition, it allows minimal instrumentation due to the simplicity of voltage application, rapid measurements (a few seconds), localized light emission, and a cost-effective setup [13]. With the substitution of an expensive detector (photon multiplier tube) and a control hardware/data collector with a cell phone camera and processor, a mobile phone-based ECL sensor has been developed for its mobility, portability, and affordability [14,15].

The ML modeling that explains the mechanism of chemical or biosensing events is relatively new to the field. Nevertheless, ML might be an essential part of the modeling arsenal with applications in the sensor technology field for detection purposes. Specifically in the field of portable ECL sensors, to the best of the authors’ knowledge, ML has been little explored with the exception of an uncommon study developed by our research team [12]. In the previous study, ML was used to model the coreactant ECL mechanism of the [Ru(bpy)_3_]^2+^/TPrA system to quantify concentrations of the luminophore [Ru(bpy)_3_]^2+^. Our current study aims to model a different mechanism; the ECL quenching mechanism of the [Ru(bpy)_3_]^2+^/TPrA system by phenolic compounds. This study proposes a strategic ML approach for the detection and quantification of specific analytes, such as phenolic compounds, to contest the inevitable challenges that a low-cost sensor would face from on-site detection.

The ECL sensor can detect phenolic compounds through the direct application of the compound in the [Ru(bpy)_3_]^2+^/TrPA system using the quenching property of phenols. The target phenol species were Vanillic acids and *p*-Coumaric acids for this study. These chemicals are effective ECL quenching agents of [Ru(bpy)_3_]^2+^, being that their monitoring is critical for the biofuel industry [16]. Currently the gold standard for measuring Vanillic and *p*-Coumaric acid is the chromatographic method, which is time consuming and expensive to operate. There is a strong need for a portable and affordable sensor that can be implemented in an industry setting.

The quenching mechanism by these phenolic compounds is complex in nature as it involves the electrochemical reaction, mass and ion transport, as well as emission. The applied potential triggers a series of reactions, including the ground state and intermediate species in the system. The excited state of the luminophore [Ru(bpy)_3_]^2+^*, which is generated from a series of redox reactions, can decay to the ground state [Ru(bpy)_3_]^2+^ without ECL emission in the presence of the oxidation products of the phenolic compounds, such as *o*- or *p*-benzoquinone. Details of the quenching mechanism have been described elsewhere [17]. The quantitative investigation of the ECL quenching mechanism to detect the concentration of involved species naturally leads to the use of partial differential equations that constitute complex mechanistic models. Alternatively, this study proposes a novel modeling approach based on ML for detection purposes.

The ECL quenching in the [Ru(bpy)_3_]^2+^/TrPA system by phenolic compounds in a mobile phone-based sensor has several common challenges that can be widely recognized among similar sensors, including: (1) nonlinear dependencies; (2) multimodal data acquisition; (3) sensor-to-sensor variations when the sensor unit is changed; (4) presence of anomalies; (5) signal fluctuations due to operational/environmental variations; (6) signal degradation by repeated use of a sensor; (7) uncertainty in finding key characteristics of a signal if newly developed; and (8) lack of available sensor data to drive generalization. In this study, we investigated whether ML strategies could efficiently overcome the familiar challenges presented by the ECL sensor, which could be applied to a variety of sensors.

This study introduces ML-based strategies to effectively analyze the nonlinear and multidimensional sensor data and overcome other challenges on a mobile phone-based, portable ECL sensor. We used various ML algorithms with complete time series data from the ECL intensity and current measured (or a combination of both called multimodal data), as well as key features extracted from the time series data. This approach can be applied to similar chemical and biosensors that potentially could be developed for practical uses in a low-cost setting. This approach can significantly advance the way data is analyzed in a practical aspect of dealing with anomalies and variability.

## 2. Materials and Methods

### 2.1. Sensor Apparatus and Measurements

Simultaneous measurements of the ECL intensity and current were carried out using a mobile phone-based ECL sensor apparatus. A consumer camera from a Samsung galaxy S10 was used to record the ECL light intensity (30 frames per second) as a movie clip. A chronoamperometry signal was recorded simultaneously (sample frequency at 1000 Hz) through a custom-made compact potentiostat. Details of the apparatus design and the potentiostat circuit operation have been described in [14,15].

Disposable screen-printed electrodes (DropSens, DRP-110) with a carbon working electrode (4 mm diameter) and Ag/AgCl reference were used. For each electrode, 15–20 quenching experiments were conducted consecutively before replacing the electrode. Due to the slight variability in electrodes, each data set was normalized based on the control experiments.

For the control experiment with zero phenolic compounds, a mixture of 1 μM of [Ru(bpy)_3_]^2+^ and 20 mM of coreactant tri-*n*-propylamine (TPrA) in 0.1 M of PBS was used. Vanillic and *p*-Coumaric acids were first dissolved in ethanol to constitute a 180 mM stock and subsequently diluted to desired concentrations of 0.1–50 μM with the solution used for the control experiments. For each electrode, 50 μL of the constituted sample was applied on the electrode. A DC voltage of −1.2 V was first applied for 1 s to establish stability and +1.2 V followed for another second to elicit the ECL reaction. A custom-built mobile app controlled the potentiostat and mobile phone camera shooting. The ECL intensity and current data were simultaneously collected in the app during the voltage application.

### 2.2. Data Preprocessing

The mobile phone-based ECL sensor simultaneously produces two types of sequential time series data (Figure 1): (1) The ECL light intensity was recorded as a movie file (mp4) by the default camera app, followed by extracting them into image sequences. The average light intensity within the region of interest (ROI) in each frame was calculated using the NIH ImageJ software. The time series of the ECL intensity data contained a very low amount of ambient light, so the baseline of each run was subtracted from the data set. The average max intensity of the control experiments of each electrode was then used to normalize the data set; (2) The electric current followed by the chronoamperometric voltage application was also recorded by a compact potentiostat in the sensor apparatus. The time series of the current data was also normalized with the average max value of the controls. The baseline was not subtracted as there was minimal contamination. The normalization preprocessing was performed because each electrode has fluctuations due to manufacturing/environmental variability and the use of scaled values often improves performance in ML training. During a 1 sec duration of applied voltage, the first 25 data points of the ECL intensity and 200 data points of the current data were used, as they were the most significant.

### 2.3. Testing Strategy

To evaluate the performance of the concentration prediction model, we used the following ML algorithms: a single, bi-layer and tri-layer neural network, SVR, Boosted Trees, and GPR. Just for comparison, a linear regression method was also used although it would not be the best choice considering non-linear dependencies of the data.

The training and test data were split using a stratified shuffle split and a 5-fold cross validation method was used to evaluate the performance more accurately. The whole data was strategically divided into 5 folds, in which 4 folds (referred to as training data) were used for training/validation of the prediction model, while 1 fold (referred to as test data) was held back for testing. Specific prediction models were trained and validated with the training data set, followed by testing with the test data for score comparison. This entire process was repeated 5 times to generate statistical metrics. Figure 2 illustrates the cycle of each process.

To compare the traditional calibration approach, data were plotted for ECL intensity vs. concentration in the range of 0–30 μM and the exponential decay fit from Excel software (Microsoft) was used.

The prediction performance of the model was evaluated through statistical metrics: R squared (R^2^), root mean squared error (RMSE), and mean absolute error (MAE). We trained the ML models with the time series data of the ECL intensity/current directly and with extracted features.

### 2.4. ML Prediction Models

#### 2.4.1. Single or Multilayer Neural Net

Typically, a neural net (NN) structure includes an input, hidden, and output layers. In this study, a single layer neural net refers to a neural net with one hidden layer, which was useful in the early age of machine learning development. Multilayer neural networks contain a series of fully connected hidden layers, which enable functioning with higher complexity and capturing higher levels of patterns. The number of neurons of the hidden layer are key parameters which affect the performance of the NN significantly [18].

For all NNs, the input layer had 225 neurons for a multimodal time series and 14 neurons for multimodal features. The single layer NN had 25 neurons in the hidden layer and the bi- and tri-layer NNs had 10 neurons in each hidden layer. For consistency purposes, the same number of neurons in the hidden layers for each bi or tri-layered NN test was used whether it was for time series data or features extracted from the time series. The output layer had one neuron for all NNs.

A Rectified Linear Unit (ReLU) was used as an activation function for it improves the training speed and performance with the computational simplicity and linear behavior. For training we used the stochastic gradient descent method that estimates the error gradient for the current state of the model using the training dataset, and then updates the model weights using the back-propagation of errors algorithm. The max iteration was limited to 1000.

#### 2.4.2. Support Vector Regression (SVR)

Support vector machines (SVMs) are designed to search for a hyperplane that maximizes the margin between the training patterns and the decision boundary. Due to their impressive performance, SVMs are a popular approach for binary classification tasks [19]. Support vector regression (SVR) is an adaptation of the SVM to regression problems. SVR uses a loss function that penalizes predicted values that fall outside an acceptable window around the true values during training. As with the SVM, SVR employs kernel methods that transform features and maps data into a higher dimensional space. The kernels allow SVR to be more flexible and able to handle nonlinear problems. For this work, the quadratic SVR method that employs 2nd order kernels was implemented.

#### 2.4.3. Trees

A decision tree is built by splitting the source set, constituting the root node of the tree, into subsets, which constitute the successor children. Trees often encounter the problem of overfitting as the tree grows deeper; ensemble trees methods are widely used for their excellent performance. As one of the ensemble methods, Boosted Trees was used. Boosting is an iterative process where models are trained in a sequential order. Boosted Trees operates by developing a number of trees to combine the output of many weak learners with weighting and applying the learners repeatedly in a series. By providing the use of simple tree learners, the Boosted Trees method is robust against overfitting problems, works well with noisy signals, and vastly improves prediction accuracy [20].

#### 2.4.4. Gaussian Process Regression (GPR)

GPR is a nonparametric, Bayesian approach to regression that works well with small datasets and provides uncertainty measurements on their prediction [21]. GPR specifies a priority on the function space, calculates a posterior using the training data, and computes the predictive posterior distribution on the points of interest. The GPR prediction is probabilistic (Gaussian) with empirical confidence intervals and provides versatility in choosing kernels. For this work, we used a square exponential kernel in the GPR method.

## 3. Results and Discussion

### 3.1. Variability from the Mobile Phone-Based ECL Sensor

The low cost, mobile phone-based ECL sensor was operated in a custom-built enclosure at room temperature [14]. The obtained sensor data may vary when the sensor units are replaced (sensor-to-sensor variations), the measurements are repeated within the same electrodes, the sensing environment fluctuates during various times of the day, and slight changes happen with operators or sample preparation. As a result, the ECL intensity and current data obtained from several electrodes showed variability even after normalization (Figure 3). The ECL intensity data from Vanillic acid has several low signals that consistently appear throughout the concentration range, while most other signals are within the norm, shown in Figure 3a. Anomalies are also seen in the *p*-Coumaric acid current data even though the ECL intensity appears more consistent among different electrodes (Figure 3b). Even after very obvious outliers or visible mistakes can be excluded in the data preprocessing by the three sigma rules, some signals appear almost like anomalies due to the imperfect sensing situations and were included in the database to reflect reality.

Overall, the signal variation for the same concentration in different electrodes can be a significant amount of up to a 64% deviation. The low-cost, quantitative point-of-care devices can have such variability and anomalies in the measurement. It is not feasible to account for the large variability using a traditional calibration between sensor measurement and concentration. The traditional calibration curve, that is, a regression equation, infers the dependent variable (analyte concentration) if it is correlated with a key feature of the system. Typically, the input features for the traditional regression are one or two at most. To quantitatively compare the ML performance with the traditional calibration approach, MAE from traditional calibration were calculated following the five-fold split method. The data were split into five groups while an average of four groups (80%) were used to generate a calibration curve and the one group (20%) was employed to report the performance. We used ECL intensity for the calibration curve because all our previous work used only ECL intensity as a sole feature for data regression [14,15,17]. For Vanillic acid, the traditional nonlinear calibration curves using the peak intensity scored an MAE of 4.50 μM with a standard deviation of 1.1 μM in the concentration prediction. The ECL intensity of the Vanillic acid contained signals that could be easily regarded as anomalies and thus, the mathematical model-based calibration shows large errors. For *p*-Coumaric, the traditional nonlinear calibration curves for a scored MAE of 2.19 μM with a standard deviation of 0.667 μM in the prediction.

The results suggest that the traditional calibration curve is not practical when data have a great deal of variability. The calibration curve can be considered an oversimplified approach to explain the complex chemical-physical meaning of the quenching mechanism of the [Ru(bpy)_3_]^2+^/TPrA system by phenolic compounds, and in this study required the predetermination of a single key feature (peak intensity) that may not have sufficient information on the system.

### 3.2. Prediction Performance of the ML Models Using Multimodal Time Series

We first investigated whether ML models could provide effective predictive performance that leverage multimodal information and mitigate problems due to data variabilities. Time series of ECL intensity and current data were used for the training, validation and testing of ML models following a five-fold cross validation, as shown in the Figure 2 schematics.

The MAE values for the prediction of Vanillic and *p*-Coumaric acids using single, bi and tri-layer neural nets, SVR, Boosted Trees, GPR, and linear regression methods were summarized in Figure 4. First, it is noted that the linear regression method, equivalent to a traditional calibration curve, performed significantly worse (higher MAE) than the ML models for both Vanillic and *p*-Coumaric acid data, indicating the nonlinear dependencies of both the intensity and current signals to the concentration of the phenolic compounds. The ML models were effective for comprehending the nonlinearity of the sensor signals to the concentration. Second, it is observed that ML using multimodal data (combined ECL intensity and current data) was effective in achieving better prediction performances. For instance, for Vanillic acid, a significantly reduced MAE was achieved using multimodal data, indicating ML can identify relationships between the intensity and current in order to infer the concentration of the phenolic compounds.

When comparing ML models, it was observed that the bi- or tri-layer neural nets using multimodal data reduced the MAE by 40–67% from the prediction results of the neural nets using a single data modality (either ECL intensity or current). This result indicated that the multilayer neural networks were effective in perceiving higher order patterns and relationships from information from two different modalities. The resulting MAE for Vanillic acid with a tri-layer neural net was at 0.86 μM, which can be interpreted as that the concentration could be approximately 0.86 μM away from the actual value if the combined multimodal data are used. Considering the significant variability in the intensity data for Vanillic acid and an MAE of 4.50 μM from a traditional calibration curve, it is an approximately 80% improvement in the prediction accuracy. The result demonstrated that the multilayer neural nets were able to account for the data variability and learn higher relationships in the combined data set.

*p*-Coumaric acid had a more consistent ECL intensity data, but there was a great disparity in the prediction of concentrations using exclusively current data (Figure 3b). All ML models showed a lower MAE using only intensity data. Using only the current data, an MAE greater than those shown by the Vanillic acid predictions was observed. Interestingly, the combined intensity and current data to infer the concentration of *p*-Coumaric acid does not show significant improvement, which could indicate a weaker relationship between the intensity and current. However, it is notable that tri-layer neural nets and Boosted Trees showed improvement in the MAE with the combined data. The resulting MAE for the prediction of *p*-Coumaric acid was of 1.69 μM and 1.74 μM with the tri-layer neural net and the Boosted Trees, respectively, using the multimodal data. Considering the test data were sampled from all electrodes and the sensor-to-sensor variation was significant, this is an excellent prediction performance. The “bigger” models, such as the tri-layer neural net and Boosted Trees were beneficial for capturing the weaker connection between intensity and current. Although bigger models are more susceptible to overfitting with small and simple datasets, in this study, for instance, increasing the number of layers could have increased the neural net’s ability to learn higher-level concepts.

The results demonstrated that ML models provide excellent prediction performance for nonlinear sensor data and successfully mitigate the large variability issues caused by sensor-to-sensor variations. In addition, ML can correlate multimodal data to provide better prediction as it can provide more information than a single modality when multilayer neural nets or Boosted Trees were used.

### 3.3. ML Regression Performance with Extracted Features

Feature extraction for chemical or biosensor often relies on researchers’ expertise, so various features were experimented on to see how they affect the ML model performance in predicting concentrations of analytes. The features for the ECL sensor time series included max intensity, slope features (linear fit on the decay curve), area underneath and polynomial features (coefficients from a quadratic fit). The model was tested with a selected subset of features of ECL intensity or current, and all multimodal features (a combination of intensity and current features) using several prediction models of single, bi-, and tri-layer neural nets, Boosted Trees, SVR, GPR and linear regression. Figure 5 shows that the prediction performance for both Vanillic and *p*-Coumaric acid significantly improved when all the features from the multimodal data were used compared to when the subset of intensity or current was used. The tri-layer neural net scored an MAE of 0.86 μM and 1.36 μM for Vanillic and *p*-Coumaric acid respectively, proving that it was able to find higher complex patterns when all the features were used.

Table 1 and Table 2 summarize the prediction performance of the ML models for multimodal time series and multimodal features for Vanillic and *p*-Coumaric acids. For Vanillic acid (Table 1), the best prediction performance was observed to be an MAE of 0.805 (R^2^ = 0.863) for multimodal time series and an MAE of 0.861 (R^2^ = 0.875) for multimodal extracted features. When multimodal data were used, the tri-layer neural network performed excellent for both time series and extracted features. This result indicated that the data processing with extracted features does not guarantee yielding improved results and time series can perform equally well with multilayer neural nets. Considering the Vanillic acid data had a great deal of variability, the multilayer neural net provided a noticeable improvement by detecting meaningful patterns from just time series.

It was interesting to see that the two compounds have distinctive characteristics in the data relationships and ML performances. For *p*-Coumaric acid, the best performance was obtained as an MAE of 1.686 (R^2^ = 0.763) for the tri-layer neural net using a multimodal time series and with an MAE of 1.051 (R^2^ = 0.864) for the GPR using extracted features. This result shows that multimodal features can improve the prediction performance while the simple combination of two time series do not improve it significantly. In both cases, the advantage of the multimodal data was clearly demonstrated.

Two notable ML models are tri-layer neural net and GPR, which produced outstanding results consistently. A common practice for providing input variables (predictors) is using features extracted from sensorgrams. In this case, the number of predictors is seven for each intensity and current signal, fourteen when combined. We have also used the preprocessed time series data as predictors, which is close to the raw data directly from experiments. The intensity had 25 and the current had 200 predictors (due to the different sampling rates). Using time series values as predictors saves significant preprocessing and feature extraction efforts. Feature engineering, developing informative features for ML algorithms, is often challenging. In practice, seemingly reasonable extracted features may not sufficiently capture the critical information needed for accurate predictions. By using the time series values from the sensor as predictors, flexible ML approaches, such as multilayer neural nets, can potentially learn their own, possibly more informative features from the signals. However, predictions based on time series values can be more difficult if the ML model is not sufficiently complex and the training set is too small. A single time series data (either from ECL intensity or current data) contain less information for prediction than a well-designed, extracted feature. Some of the time series values are simply baselines, noises are still there, and the onset could be slightly off on top of the sensor variability we mentioned earlier. Based on our result, the tri-layer neural nets and GPR were both successful at predicting concentration from time series values. This approach is particularly useful when (a) reducing preprocessing time is beneficial; (b) obvious features are not clear for a newly developed sensor; or (c) discovering unforeseen potential relationships are desired.

The proposed ML models successfully predict concentration given data collected from ECL sensors. Considering the complex nature of electrochemical reactions in the ECL quenching mechanism by phenolic compounds, it is remarkable that the ML models can achieve this from sensor time series values without extensive preprocessing and feature extraction. Developing a mechanistic or first-principle model for prediction purposes and which also explains electrochemical reactions and mass transport mechanisms on the circular electrodes is complex and time consuming. Even with such a model, there may be no guarantee of its effectiveness in a data analysis pipeline. This study demonstrates that ML models provide accurate predictions of the concentration of phenolic compounds and can account for sensor-to-sensor variations.

## 4. Conclusions

This study addresses practical challenges with the low-cost sensor devices for the detection and quantification of phenolic compounds and how to overcome the challenges with the use of powerful ML strategies. The low-cost, mobile phone-based ECL sensor generated nonlinear, multimodal data with considerable variability due to sensor-to-sensor variations and environmental fluctuations. In contrast to the traditional calibration approach, the ML models, such as tri-layer neural net or Boosted Trees, carried out effective regression tasks for detection purposes by learning higher patterns from the multimodal data. The results demonstrated that the ML models could provide a robust analysis framework for sensor data with noises and variability without extensive preprocessing. The ML analysis can compensate for the deficiencies of less stringent, simple, affordable device settings through powerful learning algorithms and thus, accelerate the implementation of low-cost sensors in a wide range of practical situations, such as the detection of phenolic compounds on-site and their monitoring in industrial environments.

## Figures and Tables

**Figure 1 sensors-21-06004-f001:**
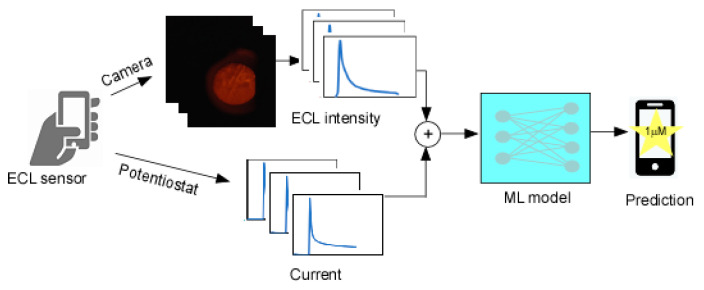
Illustration of entire process of multimodal data collection and prediction process.

**Figure 2 sensors-21-06004-f002:**
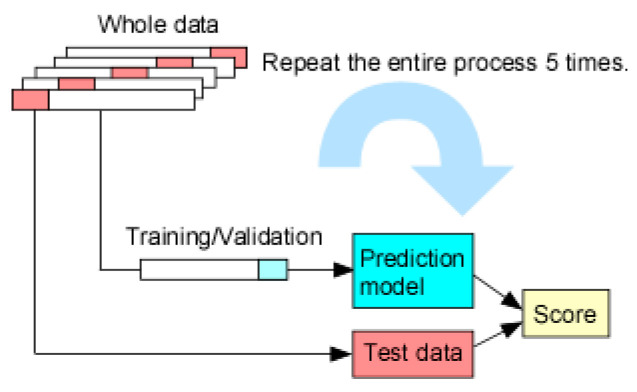
Schematics of 5-fold cross validation where the whole data is split into 5 folds. After the training/validation was completed with the training set (4 folds) in a prediction model, the test set is used to determine the accuracy of the trained model. The entire process is repeated 5 times with each split.

**Figure 3 sensors-21-06004-f003:**
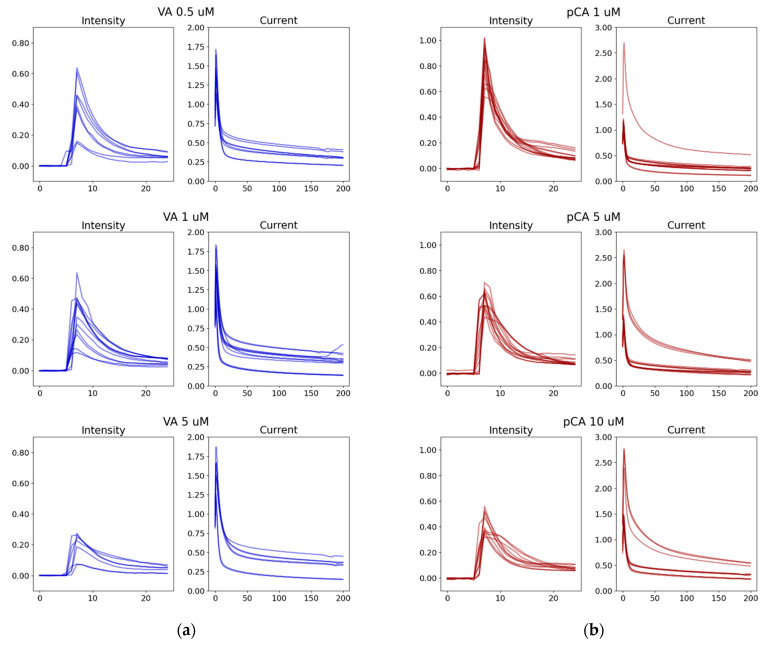
Selected examples of time sereis of ECL intensity and current from seven to eight different electrodes: (**a**) Vanillic acid (0.5, 1, and 5 μM) in blue; (**b**) *p*-Coumaric acid (1, 5, and 10 μM) in red. Horizontal axes represent measurement frame and vertical axes represent the normalized signal (arbitrary unit). These time series data were used to train ML models for Section 3.2. The data shows the variability of data that are fed into the ML algorithm. VA: Vanillic acid; pCA: *p*-Coumaric acid.

**Figure 4 sensors-21-06004-f004:**
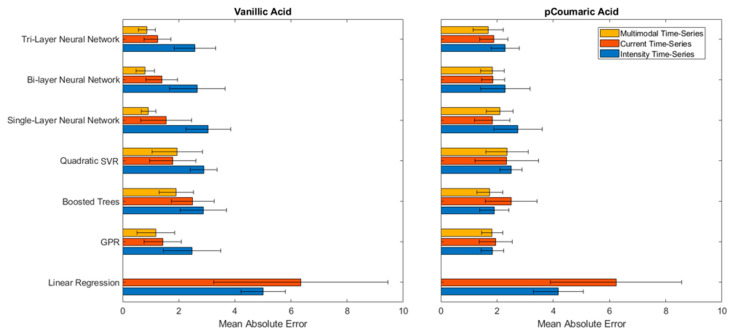
MAE test results from various ML models that were trained from the time series of multimodal (combined intensity and current), current alone, and intensity alone for Vanillic and *p*-Coumaric acids. The ML models were trained from 70 to 80 measurement experiments in the range of 0.1–30 μM and 1–50 μM for Vanillic acid and *p*-Coumaric acid, respectively. The error bars represent the standard deviation from the five-fold cross validation method. Multimodal data for Linear Regression are not provided due to the poor performance.

**Figure 5 sensors-21-06004-f005:**
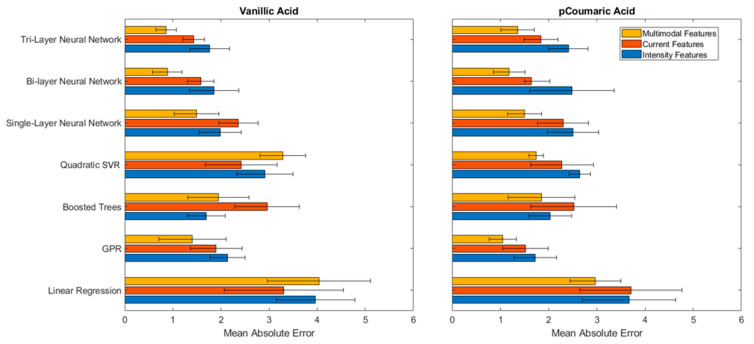
MAE of various ML models using multimodal combined extracted features versus subset of current or intensity features for Vanillic and *p*-Coumaric acids. The error bars represent the standard deviation from test results following five-fold cross validation method.

**Table 1 sensors-21-06004-t001:** ML prediction performance using multimodal time series and features for Vanillic acid.

	Multimodal Time Series			Multimodal Features		
	MAE	RMSE	R^2^	MAE	RMSE	R^2^
Tri-Layer Neural Network	0.860	1.528	0.847	0.861	1.492	0.875
Bi-Layer Neural Network	0.805	1.410	0.863	0.886	1.499	0.838
Single-Layer Neural Network	0.917	1.491	0.863	1.493	2.048	0.760
Quadratic SVR	1.946	2.724	0.688	3.287	4.805	−0.040
Boosted Trees	1.909	3.477	0.450	1.947	3.433	0.356
GPR	1.181	1.796	0.850	1.408	2.417	0.692
Linear Regression	9.503	13.601	−8.744	4.037	5.886	−0.488

**Table 2 sensors-21-06004-t002:** ML prediction performance using multimodal time series and features for *p*-Coumaric acid.

	Multimodal Time Series			Multimodal Features		
	MAE	RMSE	R^2^	MAE	RMSE	R^2^
Tri-Layer Neural Network	1.686	2.567	0.763	1.360	2.359	0.737
Bi-Layer Neural Network	1.839	2.795	0.741	1.183	1.958	0.837
Single Layer Neural Network	2.094	3.183	0.621	1.501	2.309	0.724
Quadratic SVR	2.355	3.214	0.629	1.746	2.520	0.743
Boosted Trees	1.742	2.486	0.818	1.855	2.621	0.804
GPR	1.825	2.643	0.726	1.051	1.635	0.864
Linear Regression	5.552	7.082	−1.316	2.970	3.526	0.622

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
