# Peer review of "Machine Learning Analysis for Phenolic Compound Monitoring Using a Mobile Phone-Based ECL Sensor"

_sensors, 2021, doi:10.3390/s21186004_

Round 1
Reviewer 1 Report
The authors have adopted machine learning methods to process the current and intensity signals from ECL sensors, which provides a robust mathematical model for low-cost and point-of-care detections. However, there are still some deficiencies should be modified and explained:
- Has machine learning not been applied to ECL sensor response processing in previous studies? Please clarify and provide related description.
- In Line 128, “be not be” should be “not be”.
- In Line 145, three different metrics have been mentioned for the assessment of this study. Please specify how these metrics are calculated from the outputs of the machine learning methods.
- In Section 2.4.1, please give more details of the used neural networks: a) how many neurons were set in the input and output layers, and the reasons, b) what was the activation function of the NN used in this study, c) why did you choose the mentioned numbers of neurons for the hidden layers, (d) how did you train the NNs, and (e) how did you optimize the training parameters of the NNs?
- In Section 2.4.2, why did you choose 2nd order kernel for SVM?
- Why are both classification and regression models adopted in this study. In fact, classification models output discrete class labels while regression models provide continuous values, they are used for two different situations. Please explain this.
- In Line 187-188, the authors said the response data of ECL sensors were sampled in various time and operation conditions. So, please show the detailed information of these various time and conditions.
- In Line 216, authors did not give the details of the calibration curve, which is a very important issue for the final results. As far as we know, a simple or inappropriate calibration curve naturally causes bad results. In fact, machine learning model can be also regarded as a multi-dimensional calibration curve in essence.
- In Line 217-218, the authors said “We used ECL intensity for the calibration curve because all our previous work used only ECL intensity as a sole feature for data regression”. It is unfair to set sole feature for the traditional method in comparison.
- In Figure 4, why does linear regression have only two, not three, horizontal bars?
Reviewer 2 Report
This manuscript aims to demonstrate the utilization of machine learning to overcome problems associated with conventional data analysis including liner regression (based on least squares) and other sensor variability issues. The topic is hot and but unfortunately the paper fails to distinguish itself among other ongoing research in this area. There are hundreds of papers now discussing similar findings with even more complex systems (compared to the system here that is simply has no background or any interferents see for example https://doi.org/10.1155/2014/598129; https://doi.org/10.1002/anie.201901443; and https://doi.org/10.1246/bcsj.20200359). The paper also needs to explain the mechanism of ECL quenching in Ru (bPy)3/TRPA system and the chemical principle behind this should be thoroughly discussed.
In figure 3 for example, it is very hard to tell which is what. Proper labelling of the figure and more ligands where be good for the reader to visualize and understand the point demonstrated here. It is also hard to understand what figure 4 is presenting here? The figure caption provides zero information on the analyses; what is the number of measurements, what is target concentration, what the error bars represent (some of these may be in the main text but it is critical to have them in figure caption too).
I understand that the multi-modal time series is a combination of both current and intensity. If this is true how the error associated with this multi-modal analysis is usually less than any of the other modes.
Discussion is page 9 is relatively ambiguous without providing clear conclusion and tend to just mention what has been observed without providing detailed account of why this happen or what is the trend and why.
How the paper distinguish itself from previous publication in the field. Currently there are
Specific comments.
- Line 101, why a negative reducing potential (-1.2 V) was first applied and what does establish stability means?
- Line 62, phenol spices?
- Line 17, showed comparable or better performance compared that of extracted features
- What is meant by “traditional approaches” in line 18
- Line 19 “The results demonstrated that the ML methods could provide robust analysis 19 framework for sensor data with noises and variability without”
Round 2
Reviewer 1 Report
The authors have made some modifications based on the 1st-round comments. However, there are still any issues need to be addressed:
- How do the proposed mathmatical models classifiy Vanillic acid and p-Coumaric acid ? It is impossible to predict concentration and category via one output.
- Please add the detailed information of the used calibration curve in the manuscript.
- Please replace SVM with SVR in the text, and introduce SVR in Section 2.4.2
Reviewer 2 Report
The quality of the paper has improved and can now be published in senors.
Round 3
Reviewer 1 Report
After the 2nd-round modification, I believe the quality of paper has been improved and agree the paper to be accepted by Sensors.